# Intensity and Pace Calculation of Ultra Short Race Pace Training (USRPT) in Swimming—Take-Home Messages and Statements for Swimming Coaches

**DOI:** 10.3390/sports12080227

**Published:** 2024-08-22

**Authors:** Konstantinos Papadimitriou

**Affiliations:** Faculty of Sport Sciences and Physical Education, Metropolitan College, University of East London, Thessaloniki Campus, 54624 Thessaloniki, Greece; kostakispapadim@gmail.com; Tel.: +30-698-026-5800

**Keywords:** high-intensity training in swimming, sprint intensity training in swimming, burden, physiology

## Abstract

A recently referenced method known as ultra short race pace training (USRPT), designed to familiarize swimmers with the pace of a swimming event by using high volumes and submaximal intensities, has emerged as an efficient approach, enhancing performance and predicting swimming outcomes. Despite its recognized benefits, particularly its lower physiological burden compared to other training methods, research on USRPT is still in its early stages. There are misunderstandings related to its intensity and the pace of calculation. This systematic review aims to provide valid statements identifying the pros and cons of USRPT as a training stimulus and providing swimming coaches with key messages and advice about this training method. For the analysis, 90,612 studies from PubMed, EBSCO, Science Direct, and Google Scholar databases were screened to research the background, intensity, and pace calculation of the USRPT method, although only four met the inclusion criteria. The final screening of the selected studies was conducted using a PRISMA-P document. USRPT has the potential to become a dominant training stimulus, offering a precise alternative to the often vague training sets that many swimmers use. However, further studies focusing on specific aspects of intensity and pace calculation within USRPT sets are needed for comprehensive understanding. In conclusion, USRPT appears to be a submaximal variation of high-intensity interval training (HIIT) with low blood lactate relevance to swimming events. Also, the pace calculation must be implemented considering the different demands of each point of a swimming event.

## 1. Introduction

The improvement of swimming performance demands a contribution of both aerobic and anaerobic (alactic and lactic) energy supplies [1,2] and technique [3,4]. On this basis, physiological indices such as heart rate (HR) [5,6], oxygen consumption (VO_2_, VO_2peak_, VO_2max_) [7], blood lactate (BL) [8], and blood glucose (BG) concentration [9,10], along with the subjective factor of rate of perceived exertion (RPE) of a particular session [11], are implemented to depict swimmers’ training condition before, during, and after training. The most common training types swimming coaches utilize for optimizing swimmers’ outcomes during preparation for a swimming event are divided into two main categories: (a) high-intensity and low-volume and (b) high-volume and low-intensity training. These training regimens work on both aerobic and anaerobic energy supplies [12].

### High-Intensity and Low-Volume Training Types

The duration of high-intensity interval training (HIIT), according to the American College of Sports Medicine, varies from five seconds to eight minutes, at an intensity greater than 80% of maximal heart rate (HR), or VO_2max_, and a work-to-rest ratio of 1:2 to 1:4 [12,13,14]. In swimming, the HIIT method is typically used for distances between 50 and 800 m, varying from 22 to 480 s. Atakan et al. [15] showed that the most commonly used HIIT types are submaximal efforts that elicit ≥ 90% of VO_2max_, or >75% of maximal power. Similarly, Gibala et al. [16] and Weston, Wisløff and Coombes [17] regarded HIIT as a protocol where the training stimulus is “near to maximal”, or the target intensity is between 80 and 100% of the maximal heart rate.

Based on this, authors in swimming studies have implemented bouts lasting from a few seconds to several minutes, dependent on exercise intensity, with multiple efforts interspersed by up to a few minutes of rest or less exertion (e.g., four to ten repetitions of 50 m with maximal intensity and a work-to-rest ratio near to 1:4) [10,11,17,18,19]. HIIT offers multiple performance benefits, including enhancements in both anaerobic and aerobic capacities, improvements in skeletal muscle function, and positive effects on the hematological profile [18].

Another variation of high-intensity training is sprint intensity interval training (SIIT), which involves maximal or supramaximal efforts where the intensities correspond to stimuli greater than what is required to elicit 100% of VO_2max_ [16,17,20]. Specifically, SIIT is regarded as a more intense variation of HIIT, containing bouts that last less than 30 s with long interval periods (around 4 min) [16]. The most commonly used SIIT protocol is performed on a cycle ergometer and consists of 4–6 × 30 s all-out maximal intervals, pedaling against high resistance (approximately 170% of VO_2max_), with a 4-min recovery interval or light exercise [21]. Similar training protocols have been utilized in swimming. For instance, a training set of 4 × 50 m at maximal intensity with a work-to-rest ratio of 1:4 is considered relevant based on the physiological demands of a 100 m freestyle event, and can be utilized by swimming coaches to improve their swimmers’ performance [22].

Last but not least is the repeated sprint training (RST) model, characterized by a high number of short-duration sprints (10–20 repetitions lasting less than 10 s) interspersed with brief recoveries (less than 60 s) [23]. Average oxygen consumption, as a percentage of maximal oxygen consumption (VO_2max_), ranges from 73 to 83% [24]. This training model is widely utilized in the physical preparation of athletes for many team and individual sports. In swimming, Camacho-Cardenosa et al. [25] utilized a RST program for four weeks, twice a week, consisting of 3 sets of 5 × 15 m “all out” sprints (total volume of 625 m) with 20 s of passive recovery, in both normoxic and hypoxic conditions. They demonstrated an elevation in blood lactate concentration and no difference between 100 and 400 m freestyle swimming performances.

The choice of training type depends on various factors such as the swimmers’ level (regional, national, or international), swimming style (butterfly, backstroke, breaststroke, and freestyle), swimming distance (short or long), and the period of the macrocycle (induction or racing) [11,14,19,26]. Given these training factors, coaches continually investigate the most efficient training methods for their swimmers. A recently referenced method called ultra short race pace training (USRPT), which familiarizes swimmers with the pace of a swimming event through high volumes and submaximal intensities, has emerged as an efficient way to enhance and predict swimming performance [10,27,28,29].

Despite the scarce referred benefits, indicated by the low physiological burden compared to other training methods [10,27,28,29], research on this topic is still in its early stages, with many misunderstandings related to its identity as an anaerobic stimulus, intensity, and pace calculation. Therefore, this systematic review aims to provide valid statements identifying the pros and cons of USRPT as a stimulus, orienting swimming coaches and offering key messages and advice about this type of training.

## 2. Methods

A systematic review was conducted to find the most relevant articles on USRPT. Also, the studies identification was conducted according to the PRISMA-P document [30]. The search was conducted from 7 of April until 18 July 2024 in the most strict and loose scientific databases: PubMed, SPORTDiscus, Science Direct, and Google Scholar, respectively. The main topics of the research were the background, the intensity, and the pace calculation of the USRPT method. The search strategy comprised “swimming” AND “ultra short race pace training” OR “race pace training” OR “high-intensity training” OR “sprint intensity training”. Each database’s systematic equations and PRISMA checklists are presented in Figure A1, Appendix B, and Appendix A, respectively. The studies’ eligibility was checked using the following inclusion criteria:Manuscripts in the English languageFull text availabilityHuman participantsInternational-, national-, or regional-level swimmersAcute or intervention effects of USRPT according to Rushall’s instructions:
USRPT alone or combined with other training methodsMinimum of three times for the targeted eventIntervals close to 20 s
Measurements that yielded results based on biomarkers, as well as physiological and biomechanical factorsAll study types (original, narrative reviews, systematic reviews, etc.),

While the exclusion criteria were

Not meeting the inclusion criteriaParticipants were children (≤11 years old)The volume of training was less than three times the targeted eventIntervals of less than 15 and more than 30 sStudy examined only performance factors.

## 3. Results

The literature was reviewed by examining the studies’ titles and abstracts to match the searched keywords. A total of 90,612 studies were found using the included keywords. Then, removing duplicates and articles with different content, the remaining full-text studies were selected, screened, and compared to determine inclusion in the systematic review. Twelve (12) studies were found that included in their title the keyword ultra short race pace training or USRPT; however, only four (4) met the inclusion criteria for the systematic review (PRISMA flowchart, Figure 1).

## 4. Discussion

### 4.1. Background

USRPT originated from two theories regarding brief bouts of work (sets of 25 and 50 m) [31] with rest intervals close to 20 s [32], which are considered beneficial for performance. According to Rushall [27], a typical USRPT session usually comprises short-distance bouts (15–100 m) with brief rest periods (15–25 s) and rather high volumes (up to 5–10 times the distance of the targeted event) performed at the pace of the targeted event, thus, with submaximal to maximal intensity. In these experimental studies, USRPT protocols that were utilized included 20 × 25 m on freestyle, with a 40-s interval working at a 100 m pace [10]; 20 × 50 m on freestyle, with a 1:1 interval working at a 200 m pace [29]; and 20 × 25 m on freestyle, with a 35-s interval working at a 100 m pace [28] (Table 1).

### 4.2. Intensity

The conceptualization of a USRPT protocol involves swimmers performing intervals until failure, with an increased volume of training. However, modifications in intensity probably cause swimmers to reach failure, as Rushall [27] suggests must happen in a successful USRPT set. However, Rushall’s [27] guidelines about intensity are conflicting and confusing because there is no clear statement about USRPT’s energy demands, which change depending on the distance of the event a swimmer is working on. Probably because of this, the excessive volume that a USRPT set consists of can be assumed to be aerobic-based training. However, is that all? Rushall regards exhaustion as having two main markers: (a) reduction in stored glycogen levels and (b) increment of blood lactate concentration.

Papadimitriou et al. [10] (2023); Williamson, McCarthy, and Ditroilo [28]; and Cuenca-Fernández et al. [29] examined energy demands, using blood lactate and glycogen concentration, during and after these kinds of sets, and found they largely rely on anaerobic metabolism. This can be assumed due to the blood lactate and glucose concentration that the authors examined. Specifically, Papadimitriou et al. [10] found in the first ten 25s (1–10 × 25 m) 8.7 ± 0.8 and the second ten (11–20 × 25 m) 10.0 ± 0.9 mmol·L^−1^, plus blood glucose at 1–10 × 25 m: 6.0 ± 0.9 and at 11–20 × 25 m: 6.4 ± 1.4 mmol·L^−1^. Cuenca-Fernández et al. [29] found in the first measurement at two minutes 8.2 ± 2.4 and in five minutes 6.9 ± 2.8 mmol·L^−1^, and Williamson, McCarthy, and Ditroilo [28] found 7.7 ± 2.4 mmol·L^−1^ after the first four repetitions, reaching 13.6 ± 3.1 mmol·L^−1^ at the end of the set (20 repetitions). Therefore, it is clear that energy demand, as indicated by blood lactate and glucose concentrations (>4 mmol·L^−1^), classify USRPT as an anaerobic training stimulus [33].

However, anecdotal statements suggest its potential to improve aerobic power as well [34]. Similar lactate accumulation is observed after sets oriented for strong anaerobic events such as 100–200 m. Thus, it is not established what the lactate condition would be after a USRPT set oriented for 400 or 1500 m. According to these conclusions, there are two key methods to identify USRPT as a training stimulus. The first one is to compare it with other well-established training methods (e.g., HIIT—SIIT, etc.) and conditions based on lactate concentration (e.g., onset of blood lactate accumulation [OBLA], maximal lactate steady state [MLSS], etc.). The second one is to check its relevance with specific events.

Examining these key methods, Nugent et al. [33] concluded that the physiological and perceptual demands of a typical USRPT session are similar to HIIT, with a lactate concentration above 4 mmol·L^−1^, maximal HR above 88%, and RPE values over 17. Similarly, SIIT shows common physiological demands, with a lactate concentration between 12 to 18 mmol·L^−1^ [22]; however, the anaerobic demands of a SIIT set occur with less duration and greater interval sets compared to both HIIT and USRPT.

OBLA and MLSS represent two lactate conditions at and above the second lactate threshold (≥4 mmol·L^−1^), respectively. Specifically, OBLA refers to a steady blood lactate accumulation near 4 mmol·L^−1^ [35], whereas the MLSS is the highest blood lactate concentration and workload that can be maintained under continual efforts without a continual blood lactate accumulation, indicating a balance between lactate production and its rate of clearance [36,37]. However, neither of these methods elicits a similar lactate response to a USRPT protocol; hence, it cannot be identified as a low-burden anaerobic condition, especially when considering USRPT sets only for 100 and 200 m distances. Perhaps USRPT sets for longer events (>400 m) elicit a lower anaerobic response akin to OBLA and MLSS conditions.

The second method involves assessing the relevance of USRPT to targeted events. Terzi et al. [22] demonstrated a high relevance between a 100 m event and a SIIT set oriented to 100 m, comprising 4 × 50 m repetitions, but a similar study focusing on USRPT relevance has not been conducted yet. Avlonitou [38]; Schnitzler, Seifert, and Chollet [39]; Vescovi, Falenchuk, and Wells [40]; Sousa et al. [41]; and Zacca et al. [42] have documented blood lactate patterns across various race distances (50, 100, 200, 400, 800, and 1500 m events) in swimmers of different levels and ages. These lactate values can serve as guidance for authors to construct more specific USRPT sets tailored to the physiological demands of each event and to better understand the potential contribution of this training method (Table 2).

Of course, this comparison cannot definitively establish the relevance of USRPT sets to specific targeted events due to the variations in levels, sex, and ages of swimmers in these studies. Additionally, only the freestyle stroke has been studied, and peak blood lactate concentrations were found at different time points. Therefore, researchers should focus their studies on elucidating the relevance of USRPT to targeted events, examining biochemical (BL, BG, etc.), ergophysiological (VO_2max_, VO_peak_, etc.), and swimming efficiency [10] indices. Williamson, McCarthy, and Ditroilo [28] were the only researchers to find a similar blood lactate response between a USRPT set and the 100 m freestyle event, supporting Rushall’s theory; however, as previously described, there are differences among the studied samples.

Take-home messages and statements are suggested:

USRPT is demonstrated as a highly demanding anaerobic training stimulus, hypothesized based on blood lactate concentration and swimmers reaching failure, mainly due to depleting glycogen supplies. However, there is a lack of studies regarding blood lactate relevant to the targeted event. Nevertheless, coaches are more inclined to use USRPT as a training set for distances between 400 and 10,000 m, especially for open-water swimmers, as steady pacing and blood lactate concentration are more frequently encountered over these distances. On the other hand, for sprint distances between 50 and 200 m, where the anaerobic contribution is even higher, it is more appropriate to utilize HIIT and SIIT sets with longer intervals. Lastly, for younger swimmers with fewer anaerobic demands (≤4 mmol·L^−1^), USRPT can be a beneficial method for distances of 50 and 100 m, focusing on shorter paces of 15 or 25 m and thereby familiarizing them with set tests or challenges.

### 4.3. Pace Calculation

A central contributor to constructing a successful USRPT set is pace calculation. Rushall regards the calculation of such sets to include even the first split, starting from the block, of an event. The advantage of the dive is typically included in calculating the repetition time, which means the training pace for surface swimming is slightly faster than the actual race pace from which it is calculated [43]. However, is it correct to calculate the pace based on the first split of a race? A more accurate approach is to calculate the pace of an event without considering the first 25, 50, or 100 m of it (depending on the distance). Table 3 demonstrates how a coach can accurately calculate the pace for a 200 m event by averaging the pace between the second and fourth splits. Specifically, the magnitude of difference between the average pace of the entire event (including the first 50 m) and the first split is greater compared to the average pace from the second to the fourth split.

Therefore, with this accurate approach to calculation, perhaps the physiological burden would be more specific, aligning with the event’s demands. However, Rushall supports the notion that calculating the first part of an event in a USRPT set provides an inherent “improvement factor” which should lead to continual race improvements. This sounds reasonable, considering the principle of gradual incremental training load. However, if a swimmer has a fast or slow first split, this method introduces bias, potentially making a USRPT set more difficult or easier to implement.

Another crucial point is the pace of 100 m events, because the second and fourth splits do not benefit from a push-off from the wall. In this scenario, there is no specific instruction [27,43,44]; thus, it is suggested that a coach should calculate the average pace based on the entire distance. Or is it more appropriate to calculate the average pace separately for the first and third 25 m (with a wall push-off), and the second and fourth 25 m (without a wall push-off)? Table 4 provides two possible scenarios. In the first scenario, the coach implements USRPT by calculating the average pace of a 100 m event, whereas in the second scenario, performance is calculated using splits with and without a wall push-off.

In the literature, none of the authors who implemented USRPT [10,27,28,29] seem to have considered any specific calculation of the pace according to the above-mentioned statements in sets oriented for 100 and 200 m events.

Take-home messages and statements are suggested:

Prefer the construction of a USRPT set close to the average of an event without the contribution of the first split, utilizing different training stimuli for improvement of the first split (HIIT or SIIT). Also, on occasions when an event includes splits with and without a wall, it is more appropriate to construct a USRPT set fitted to the demands of the race strategy, improving specific points of an event, such as a slow fourth split in 100 m. Therefore, set a pace according to the performance that needs to be worked on for the swimmers’ improvement.

## 5. Conclusions

USRPT has the potential to dominate as a training stimulus, avoiding the useful yet imprecise training sets that many swimmers implement. However, further studies on USRPT aspects such as intensity and pace calculation should be investigated in depth. According to the literature, USRPT seems to be a variation of HIIT without high relevance between the physiological burden of a swimming event and the respective USRPT set. Further research will enhance the precision of constructing such sets and the benefits of USRPT’s physiological and coaching potential. Additionally, the interval and volume factors of USRPT must be thoroughly discussed and investigated to fully understand this training method.

## Figures and Tables

**Figure 1 sports-12-00227-f001:**
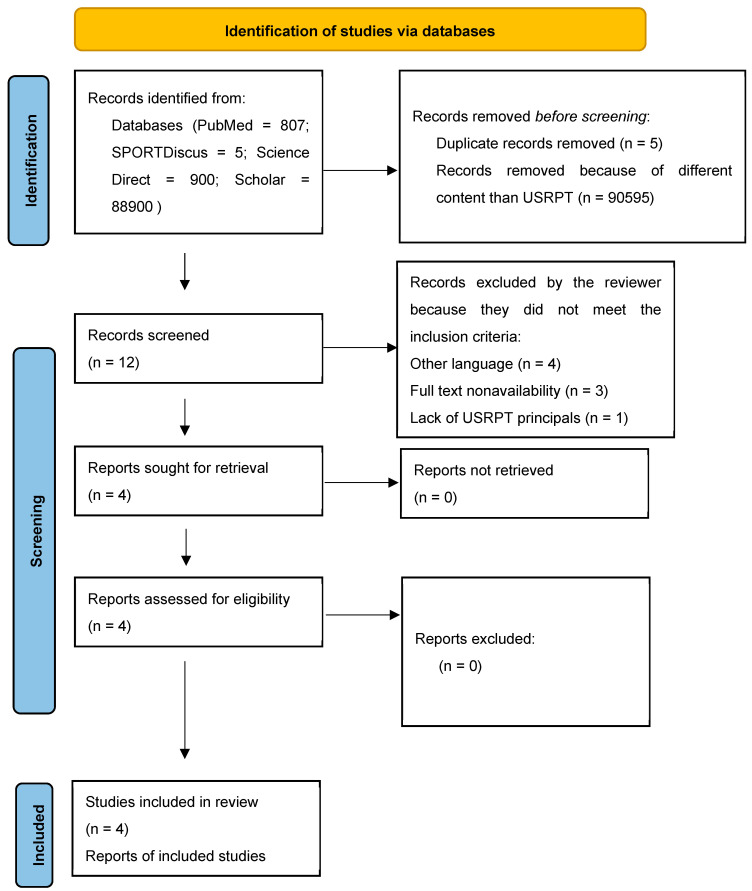
PRISMA flow chart. USRPT: ultra short race pace training.

**Table 1 sports-12-00227-t001:** Experimental studies on USRPT protocols.

Authors	Participants	Training Content	Variables Studied	Results		Limitations	Key Message
Papadimitriou et al. [10]	18 swimmers (8 ♂ and 10 ♀) 13.5 ± 0.1 years WA points > 500	USRPT: 20 × 25 m front crawl @40 s on 100 m pace HIIT: 5 × 50 m front crawl @3 min	Acute response and comparison of BL, BG, DPS, SR, SV, SI, HR, RPE	HIIT DPS ↑ SI ↑ BL ↑ HR ↑	USRPT SV ↑	Did not study the physiological response and kinematics solely for a 100 m front crawl Acute response	USRPT is an anaerobic training stimulus with a lower physiological burden than HIIT.
Williamson, McCarthy, and Ditroilo [28]	14 swimmers (7 ♂ and 7 ♀) 20.0 ± 1.6 years Elite and Sub elite swimmers	USRPT: 20 × 25 m front crawl @35 s on 100 m pace	Acute response of BL, HR, RPE	USRPT BL ↑ HR ↑ RPE ↑		Lack of comparison with other training methods Did not study the physiological response and kinematics of a 100 m front crawl Did not study the kinematics of the USRPT set Many intervals during the USRPT protocol for the measurement of BL concentration Acute response	USRPT, according to HR, BL, and RPE indices, is an anaerobic training stimulus.
Cuenca-Fernández et al. [29]	14 swimmers (? ♂ and ? ♀) 19.0 ± 1.6 and 19.0 ± 0.1 years WA points > 500	USRPT: 20 × 50 m front crawl @60 & 70 s for ♂ and ♀, on 200 m pace RPT: 10 × 100 m front crawl @130 & 140 s for ♂ and ♀, on 200 m pace	Acute response of BL, SC, CMJ	RPT BL ↑ SC ↑ CMJ ↓ Compared to USRPT		More physiological measurements during the two protocols Did not study the physiological response or kinematics of a 100 m front crawl Acute response	USRPT has a lower metabolic burden than RPT. RPT could be a training method for greater distance swimming events.

WA = World Aquatics; ♂ = males; ♀ = females; USRPT = ultra short race pace training; HIIT = high intensity interval training; BL = blood lactate; BG = blood glucose; DPS = distance per stroke; SR = stroke rate; SV = swimming velocity; SI = stroke index; HR = heart rate; RPE = rate of perceived exertion; ↑ = increased response; ? = not defined; RPT = race pace training; SC = stroke count; CMJ = counter movement jump; ↓ = decreased response.

**Table 2 sports-12-00227-t002:** Relevance of blood lactate response between swimming events (left columns) and USRPT sets (right columns) oriented to 100 and 200 m.

	Distances (m)				
BL in Events (mmol·L^−1^)	100		200		BL in USRPT (mmol·L^−1^)
Avlonitou [38]	13.1 ± 2.7	**20 × 25 m, 13.6 ± 3.1**	12.8 ± 1.3		Williamson, McCarthy, & Ditroilo [28]
Vescovi, Falenchuk and Wells [40]	13.9 ± 1.9		14.0 ± 1.7	20 × 50 m 8.2 ± 2.4	Cuenca-Fernández et al. [29]
Sousa et al. [41]		20 × 25 m, 10.0 ± 0.9	11.7 ± 1.4		Papadimitriou et al. [10]
Zacca et al. [42]	12.4 ± 1.8		12. ± 1.6		

BL: blood lactate response; USRPT: ultra short race pace training; Bold: Relevance between swimming event and USRPT.

**Table 3 sports-12-00227-t003:** Demonstration of a specific approach to construct an accurate USRPT set independently of the tactic that a swimmer follows at the first split.

200 m Event on 120 s	First Split (s)	Second Split (s)	Third Split (s)	Fourth Split (s)	Av. Pace of Total Event (s)	Av. Pace of Second to Fourth Split (s)	Df. between Av. Pace of Total Event and First Split (s)	Df. between Av. Pace of the Total Event and Av. Pace of Second to Fourth Splits (s)
The hurrying swimmer	27	30	32	31	30	31	−3	−1
The patient swimmer	28	30	32	30	30	30.6	−1	−0.4
The negative swimmer	30	31	30	29	30	30	0	0

Av.: average; Df.: difference.

**Table 4 sports-12-00227-t004:** Demonstration of a specific approach to construct an accurate USRPT set oriented to 100 m, according to splits with or without the wall push-off.

100 m Event (s)	First Split (s)	Second Split (s)	Third Split (s)	Fourth Split (s)	Av. Pace of Total Event (s)	Av. Pace of Wall Splits (First—Third) (s)	Av. Pace without Wall Splits (Second—Fourth) (s)	Df. between Occasions (s)
60	13	16	14	17	15	13.5	16.5	−1.5

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
