# Peer review of "Intensity and Pace Calculation of Ultra Short Race Pace Training (USRPT) in Swimming—Take-Home Messages and Statements for Swimming Coaches"

_sports, 2024, doi:10.3390/sports12080227_

Round 1

Reviewer 1 Report (Previous Reviewer 2)

Comments and Suggestions for Authors

Dear authors, 

After revising the paper titled: Intensity and Pace Calculation of Ultra Short Race Pace Training (USRPT) in Swimming. Take-home Messages and Statements for Swimming Coaches

I consider your paper to be published in Sports (MDPI)

Comments on the Quality of English Language

English is fine, minor improvements are required.

Author Response

Dear Reviewer

Thank you for the acceptance of my manuscript. I provided the necessary improvements in the English language.

Kind regards

Reviewer 2 Report (Previous Reviewer 4)

Comments and Suggestions for Authors

The authors of this publication “Intensity and Pace Calculation of Ultra Short Race Pace Training (USRPT) in Swimming: Take-home Messages and Statements for Swimming Coaches” has considered most of my remarks made during my first review process. Nevertheless, in my opinion, there are some points to must be fixed additionally:

1.              Keywords below the abstract in my opinion as well could be expanded with inclusion of “swimming”.

2.              Moreover, it would be valuable to present examples of competitive activities in swimming which last “few seconds” (lines 48-49) was not presented in the article.

Comments on the Quality of English Language

The authors of this publication “Intensity and Pace Calculation of Ultra Short Race Pace Training (USRPT) in Swimming: Take-home Messages and Statements for Swimming Coaches” has considered most of my remarks made during my first review process. Nevertheless, in my opinion, there are some points to must be fixed additionally:

1.              Keywords below the abstract in my opinion as well could be expanded with inclusion of “swimming”.

2.              Moreover, it would be valuable to present examples of competitive activities in swimming which last “few seconds” (lines 48-49) was not presented in the article.

Author Response

Dear Reviewer

Thank you for the comments.

29-30. I included the keyword "swimming" next to high and sprint-intensity training.

49-50. I included a sentence guiding the readers about the racing distances which are similar to a HIIT protocol.

Kind regards

This manuscript is a resubmission of an earlier submission. The following is a list of the peer review reports and author responses from that submission.

Round 1

Reviewer 1 Report

Comments and Suggestions for Authors

I recommend accepting the article. 

Author Response

Thank you for the kind feedback!

Reviewer 2 Report

Comments and Suggestions for Authors

Dear authors, 

Your article seems very suitable and relevant in the field.

However, I perceive a need to include more databases to make the search deeper and more representative of the entire literature. Furthermore, the Google Scholar database is not the best option when focusing on retrieving a greater level of evidence.

Finally, in the next version of your paper, please add an appendix with the hole systematic equations for each database.

After these improvements, I will consider your systematic review for a second round of reviews.

No further comments.

Best regards. 

Comments on the Quality of English Language

English is fine. 

Minor reviews are required in the last version in case the paper is accepted for publication.

Author Response

Dear Reviewer 2. Thank you very much for your insightful comments and the belief that this article is relevant to the field.

C: Comment

A: Answer

C: I perceive a need to include more databases to make the search deeper and more representative of the entire literature.

A: I agree with you. As you suggested, SPORTDiscus and Science Direct databases were included, considering that these databases found relevant articles about USRPT. Lines 102-103.

C: Furthermore, the Google Scholar database is not the best option when focusing on retrieving a greater level of evidence.

A: It is true that Google Scholar includes articles with many biases and, in some cases, low impact. However, as you probably understand, the present topic lacks sufficient scientific literature. In my opinion, a topic with limited scientific evidence requires a narrative review as a first step in the analysis. Of course, systematic reviews have a greater scientific impact than narrative reviews; however, this topic has only three original articles and many unanswered questions, especially considering George Rushall’s theory, which is supported by unpublished data. In the present manuscript, I aim to provide a source for authors to develop new methodologies on USRPT.

C: Finally, in the next version of your paper, please add an appendix with the whole systematic equations for each database.

A: As you suggested, I included a table demonstrating the equitation and keywords utilized to search articles. Lines 563 – 557.

C: After these improvements, I will consider your systematic review for a second round of reviews.

A: I am glad to answer your further comments. Thank you. 

Reviewer 3 Report

Comments and Suggestions for Authors

Abstract:

- In the methodology place the total number of studies analysed.

Introduction:

- Lines 34-38: Refer to this statement.

- Lines 40-54: Relate this paragraph or clarify why it is related to your study. Otherwise, together with the following paragraph (lines 56-61).

Methods:

- Authors are encouraged to begin this section by explaining the type of study they conducted (systematic review).

- There are very few studies included... The inclusion or exclusion criteria may need to be revised. Or, failing that, focus the review on other variables or training methods related to USRPT.

Results:

- It is strongly recommended that the authors include a results section with the results tables and a description of the results.

Discussion:

- It is strongly recommended to the authors to add a discussion section in which the results extracted in the study are compared with the results of other research in order to arrive at coherent statements that help and contribute to science.

I recommend a section on practical applications.

Author Response

Dear Reviewer 3. Thank you very much for your comments. I checked them in detail and I hope my answers satisfy you.

C: Comment

A: Answer

Abstract:

C: In the methodology place the total number of studies analysed.

A: I included the total number of studies and the final number of analyzed. Lines: 17-20

For the analysis, 90612 studies from PubMed, EBSCO, Science Direct, and Google Scholar databases were screened to research the background, intensity, and pace calculation of the USRPT method, whereas only four met the inclusion criteria.

Introduction:

C: Lines 34-38: Refer to this statement.

A: I included the reference for this statement. Line 41

C: Lines 40-54: Relate this paragraph or clarify why it is related to your study. Otherwise, together with the following paragraph (lines 56-61).

A: The structure of the introduction includes a general overview of the components of HIIT (Lines 45-52), followed by a description of its implementation in swimming (Lines 53-58). I apply the same approach for SIT and RST, demonstrating the variations of high-intensity training. This leads to a conclusion on USRPT, which, according to the analysis, is also a variation of high-intensity training.

Methods:

C: Authors are encouraged to begin this section by explaining the type of study they conducted (systematic review).

A: As you correctly suggested, I included the type of the study, identifying that is a systematic review. Lines 99-101

A systematic review was conducted to find the most relevant articles on USRPT. Also, the methodology studies identification was conducted according to the PRISMA-P document [30].

C: There are very few studies included... The inclusion or exclusion criteria may need to be revised. Or, failing that, focus the review on other variables or training methods related to USRPT.

A: I understand your point of view. USRPT is a new type of high-intensity training that likely offers many benefits. However, at present, there are only three original articles examining its acute effects and one systematic review by Nugent, F., Comyns, T., Kearney, P., & Warrington, G. (2019) titled "Ultra-Short Race-Pace Training (USRPT) In Swimming: Current Perspectives" published in the Open Access Journal of Sports Medicine. According to Nugent et al., “all studies were excluded as the intervention did not consist of USRPT. Consequently, there are concerns surrounding USRPT as it is not currently based on peer-reviewed published literature.”

In my opinion, due to the limited scientific evidence available, a narrative review is necessary as a first step in the analysis. While systematic reviews typically have a greater scientific impact than narrative reviews, this topic only has three original articles and many unanswered questions, especially considering George Rushall’s theory, which is supported by unpublished data. Also, the exclusion criteria have a minimum of requirements:

  1. Manuscripts in English language
  2. Full text availability
  3. Human participants
  4. International or National or Regional level swimmers
  5. Acute or intervention effects of USRPT, according to Rushall’s instructions
    1. USRPT, as a unique or combined with other training method
    2. Minimum of three times the targeted event
  • Interval close to 20 sec
  1. The measurements extract results accordingly with biomarkers, physiological and biomechanical factors
  2. All study types (Original, Narrative Reviews, Systematic Reviews, etc.)

These requirements contribute to having specific content on USRPT utilizing other training methods for comparison.

Results:

C: It is strongly recommended that the authors include a results section with the results tables and a description of the results.

A: I included the PRISMA flow chart in the results section to provide a detailed description of the number of studies that were screened and ultimately included. Lines 138-176.

C: It is strongly recommended to the authors to add a discussion section in which the results extracted in the study are compared with the results of other research in order to arrive at coherent statements that help and contribute to science.

A: According to your insightful comments, the discussion section includes all the extracted results which are further discussed with other training methods and studies, concluding with “Take home messages and statements. Lines 178 – 334.

C: I recommend a section on practical applications.

A: With all due respect, the present journal does not include a dedicated practical application section. However, the "Take Home Messages and Statements" section should satisfy your comment, as it serves as the practical application of the study. This section provides readers with insightful information on USRPT and its application. Lines 273-283 & 328-337.

Reviewer 4 Report

Comments and Suggestions for Authors

The article “Intensity and Pace Calculation of Ultra Short Race Pace Training (USRPT) in Swimming: Take-home Messages and Statements for Swimming CoachesThe paper provides a systematic review focused on Ultra Short Race Pace Training (USRPT) in swimming, analyzing its intensity and pace calculations. The study stands out for addressing the current gaps in understanding USRPT, particularly its physiological burden and effectiveness as a training method. The methodology, using PRISMA-P guidelines and databases like PubMed and Google Scholar, adds a robust framework to the review.

There are several areas in this study raises critical points.  could be improved for greater scientific rigor and applicability:

1.              The review includes only a small subset of available studies, with only four studies meeting the inclusion criteria. This narrow selection may limit the generalizability of the findings. In addition, in table 1 there is presented data just from 3 research. It is not clear why.

2.              The review includes only a small subset of available studies, with only four studies meeting the inclusion criteria. This narrow selection may limit the generalizability of the findings. It would be valuable to include data analysis with effect size and Cohen D calculation etc.

3.              Keywords could be expanded with inclusion of “swimming”.

4.              While the discussion outlines physiological responses to USRPT, it does not sufficiently compare these responses to those elicited by other training methods like HIIT in terms of specific physiological markers (e.g., VO2 max, lactate threshold).

5.              Moreover, it would be valuable to present examples of competitive activities which last “few seconds” (lines 48-49).

6.              In addition, in Materials and Methods part there are not clear presented research exclusion criteria.

7.              In Figure 1 presented some remarks that are not explained (e.g. “*”).

8.              Notes below the tables are not presented or not presented clearly (e.g. 1, 2 and 3 table). Why some data presented in bold?

9.              Research results part and discussion part presented in one, therefore it is very complicated to understand and separate research data and the discussion. In addition, it is not clear from there was taken data in table 3 and 4.

10.           The paper provides generalized recommendations for swimming coaches but does not delve into how these can be implemented in varied coaching environments or adjusted based on the swimmers' competitive levels, age, sex etc.

11.           In the research there is plenty of abbreviations used that are not explained (e.g. SIIT etc.).

12.           The numbering of pages in the article is not correct.

13.           References in the article were presented not in accordance with the article’s presentation requirements. Moreover, the majority of references are quiet old (25 of 44 published before 2019 year) even some of them from previous century (e.g. 26, 31 34, 35 etc.).

Comments on the Quality of English Language

The article “Intensity and Pace Calculation of Ultra Short Race Pace Training (USRPT) in Swimming: Take-home Messages and Statements for Swimming CoachesThe paper provides a systematic review focused on Ultra Short Race Pace Training (USRPT) in swimming, analyzing its intensity and pace calculations. The study stands out for addressing the current gaps in understanding USRPT, particularly its physiological burden and effectiveness as a training method. The methodology, using PRISMA-P guidelines and databases like PubMed and Google Scholar, adds a robust framework to the review.

There are several areas in this study raises critical points.  could be improved for greater scientific rigor and applicability:

1.              The review includes only a small subset of available studies, with only four studies meeting the inclusion criteria. This narrow selection may limit the generalizability of the findings. In addition, in table 1 there is presented data just from 3 research. It is not clear why.

2.              The review includes only a small subset of available studies, with only four studies meeting the inclusion criteria. This narrow selection may limit the generalizability of the findings. It would be valuable to include data analysis with effect size and Cohen D calculation etc.

3.              Keywords could be expanded with inclusion of “swimming”.

4.              While the discussion outlines physiological responses to USRPT, it does not sufficiently compare these responses to those elicited by other training methods like HIIT in terms of specific physiological markers (e.g., VO2 max, lactate threshold).

5.              Moreover, it would be valuable to present examples of competitive activities which last “few seconds” (lines 48-49).

6.              In addition, in Materials and Methods part there are not clear presented research exclusion criteria.

7.              In Figure 1 presented some remarks that are not explained (e.g. “*”).

8.              Notes below the tables are not presented or not presented clearly (e.g. 1, 2 and 3 table). Why some data presented in bold?

9.              Research results part and discussion part presented in one, therefore it is very complicated to understand and separate research data and the discussion. In addition, it is not clear from there was taken data in table 3 and 4.

10.           The paper provides generalized recommendations for swimming coaches but does not delve into how these can be implemented in varied coaching environments or adjusted based on the swimmers' competitive levels, age, sex etc.

11.           In the research there is plenty of abbreviations used that are not explained (e.g. SIIT etc.).

12.           The numbering of pages in the article is not correct.

13.           References in the article were presented not in accordance with the article’s presentation requirements. Moreover, the majority of references are quiet old (25 of 44 published before 2019 year) even some of them from previous century (e.g. 26, 31 34, 35 etc.).

Author Response

Dear Reviewer 4. Thank you very much for your comments. I checked them in detail and I hope my answers satisfy you.

C: Comment

A: Answer

  1. The review includes only a small subset of available studies, with only four studies meeting the inclusion criteria. This narrow selection may limit the generalizability of the findings. In addition, in table 1 there is presented data just from 3 research. It is not clear why.

A: I understand your point of view. USRPT is a new type of high-intensity training that likely offers many benefits. However, at present, there are only three original articles examining its acute effects and one systematic review by Nugent, F., Comyns, T., Kearney, P., & Warrington, G. (2019) titled "Ultra-Short Race-Pace Training (USRPT) In Swimming: Current Perspectives" published in the Open Access Journal of Sports Medicine.

Therefore, I decided to include only the original articles in Table 1 because the review covers the same articles. According to Nugent et al., “all studies were excluded as the intervention did not consist of USRPT. Consequently, there are concerns surrounding USRPT as it is not currently based on peer-reviewed published literature.

In my opinion, due to the limited scientific evidence available, a narrative review is necessary as a first step in the analysis. While systematic reviews typically have a greater scientific impact than narrative reviews, this topic only has three original articles and many unanswered questions, especially considering George Rushall’s theory, which is supported by unpublished data.

Additionally, my article primarily presents hypotheses and suggestions for the methodological development of this training method. Therefore, I avoided making absolute statements and explanations, recognizing USRPT’s immature condition.

C: The review includes only a small subset of available studies, with only four studies meeting the inclusion criteria. This narrow selection may limit the generalizability of the findings. It would be valuable to include data analysis with effect size and Cohen D calculation etc.

  1. As I mentioned in the previous comment, I avoided making absolute statements and explanations, recognizing the immature state of USRPT. Additionally, while calculating Cohen’s d is a brilliant suggestion, this is a systematic review and not a meta-analysis. The purpose of this review is to provide suggestions for further studies. I believe that this topic is better suited for a narrative review. Conducting a systematic review and meta-analysis could be easily implemented if a minimum of 10 studies on this topic existed.

C: Keywords could be expanded with inclusion of “swimming”.

A: Thank you for the comment. If you mean the keywords in the abstract, I deliberately avoided using keywords similar to those in the title. However, in the methodology section, I specified that the first keyword in the research strategy was “swimming. ”Lines: 104-106

The search strategy comprised “swimming” AND “ultra-short race-pace training” OR “race pace training” OR “high-intensity training” OR “sprint intensity training”.

C: While the discussion outlines physiological responses to USRPT, it does not sufficiently compare these responses to those elicited by other training methods like HIIT in terms of specific physiological markers (e.g., VO2 max, lactate threshold).

A: In lines 207–218, I describe the physiological responses to USRPT. From lines 214–254, I compare USRPT with HIIT, SIT, OBLA, and MLSS methods in terms of blood lactate, RPE, glucose, and heart rate. The main goal is to compare the indices studied in both USRPT and other variations of high-intensity training to help readers better understand USRPT’s physiological burden.

Moreover, it would be valuable to present examples of competitive activities that last “few seconds” (lines 48-49).

C: In addition, in Materials and Methods part there are not clear presented research exclusion criteria.

A: According to your comment, I modified the exclusion criteria, specifying some of them and keeping the “Not the inclusion”. Lines 122-127.

  1. Not the inclusion
  2. Participants were children (≤ 11 years old)
  3. The volume of training is less than three times the targeted event
  4. Interval less than 15 and more than 30 sec
  5. Studied only performance factors

C: In Figure 1 presented some remarks that are not explained (e.g. “*”).

A: Figure 1 is the PRISMA flowchart. I included the explanation of USRPT and I deleted (*).

C: Notes below the tables are not presented or not presented clearly (e.g. 1, 2 and 3 table). Why some data presented in bold?

A: Thank you for your comment. After an extended check: In Table 1, I removed Bold words and I explained WA = World Aquatics, and the symbols of Males and Females (195-199). In Table 2, I explain that Bold is used for the relevance of Blood Lactate concentrations (Lines: 262 – 263). In Table 3 all the abbreviations have been explained (Line 305).

C: Research results part and discussion part presented in one, therefore it is very complicated to understand and separate research data and the discussion. In addition, it is not clear from there was taken data in table 3 and 4.

A: I understand your point. I separated the Results and Discussion sections. Results (Lines: 137 – 180) and Discussion (Lines: 182 – 338).

C: The paper provides generalized recommendations for swimming coaches but does not delve into how these can be implemented in varied coaching environments or adjusted based on the swimmers' competitive levels, age, sex etc.

A: I understand the importance of your comment. This training method still has many misconceptions, and there is no scientific guidance for implementing USRPT regarding factors like sex, level, and age. Therefore, I have noted in the manuscript that the conclusions cannot be definitively interpreted. Specifically, in lines 265–273, I outline the limitations regarding the relevance of USRPT compared to other training methods. Furthermore, the take-home messages are intended to be advisory rather than definitive, as the data do not support precise statements.

In lines 307–321, I offer suggestions on how a swimming coach can calculate the pace for this training method. In conclusion, as I mentioned previously, the purpose of this review is to explore new aspects of USRPT that will contribute to enhancing this method and other high-intensity variations.

C: In the research there is plenty of abbreviations used that are not explained (e.g. SIIT etc.)

A: I checked again the abbreviations providing corrections in the whole manuscript.

C: The numbering of pages in the article is not correct.

A: Thank you for the detail. This is the form of MDPI Sports, therefore, it is difficult to edit it. If the article is accepted, I believe that the editorial office will provide the last details.

C: References in the article were presented not in accordance with the article’s presentation requirements.

A: According to the instructions for authors, MDPI provides free format submission. However, I checked the references in the manuscript and I modified them.

C: Moreover, the majority of references are quiet old (25 of 44 published before 2019 year) even some of them from previous century (e.g. 26, 31 34, 35 etc.).

A: I understand your point. In some cases, it is important to describe basic terms by using foundational references. Specifically:

Ref 26: Justifies the 4-mmol/l lactate concentration. This information has been used in the introduction.

Ref 31: This article was the basis that George Rushall used for the construction of USRPT.

Ref 34: Provides foundational information about aerobic power.

Ref 35: Discusses the effects of OBLA accumulation.

In general, the reason I have used some older references is that they precisely describe the concepts I want to address in the manuscript. Additionally, many recent articles reference these foundational works in their introductions.